# Using Non-Violent Discipline Tools: Evidence Suggesting the Importance of Attunement

**DOI:** 10.3390/ijerph20247187

**Published:** 2023-12-15

**Authors:** Karen R. Quail, Catherine L. Ward

**Affiliations:** Department of Psychology, Faculty of Humanities, University of Cape Town, Cape Town 7700, South Africa; catherine.ward@uct.ac.za

**Keywords:** attunement, attachment, parenting, classroom management, discipline, behavior management

## Abstract

Training in non-violent discipline is important to prevent violence against children and ensure that their caregivers remain a safe base for them. This paper aims to deepen understanding of non-violent discipline by exploring attunement as a mechanism in the effectiveness of non-violent discipline tools. Attunement describes the sensitive responsiveness of caregivers towards their children and has been found to be central to the formation of secure attachment bonds and development of self-regulation. It includes understanding or being “in tune with” the child’s needs and signals, matching these with appropriate responses. The objective of this paper is to explore attunement in relation to non-violent discipline. Peer-reviewed systematic reviews previously included in a systematic overview of evidence on non-violent discipline options were screened for information relevant to attunement. All reviews were published in English between 1999 and 2018 and offered evidence on at least one non-violent discipline tool. Although no reviews explicitly addressed attunement, evidence was found suggesting its importance in the use and effectiveness of discipline methods. Research directly investigating attunement in discipline is needed.

## 1. Introduction

### 1.1. Non-Violent Discipline

Coercive and violent approaches to child discipline, such as threats, humiliation and corporal punishment, are not only abusive and damaging in themselves [1,2,3], but have long been known to foster aggression and conduct problems, and predict poorer mental health in adolescence and adulthood [4,5,6,7,8,9,10,11]. On the other hand, permissive parenting has negative effects. When parents do not hold appropriate boundaries, their children tend to show lower self-control, poorer social skills, and aggression, as well as mental health and relational problems reaching into adolescence and adulthood [12,13,14,15]. Caregiver use of effective non-violent discipline methods could, therefore, be expected not only to improve child wellbeing, but also the wellbeing and safety of others as these children reach adolescence and adulthood.

Since the word *discipline* is often associated or used synonymously with the word *punishment*, it is important to clarify its use in this paper. Discipline is defined here as constructively addressing a child’s resistance, lack of co-operation or problem behavior, and, also, as teaching and supporting appropriate behavior [16]. This support and guidance offered by adults should promote eventual self-discipline on the part of the child [12,13,17,18,19,20,21,22]. The behaviorist definition of *punishment* includes any change in the environment which results in a behavior being less likely to be repeated [23]. This definition indicates a partial fit with the goals of discipline; however, in common use, the word punishment often describes inflicting physical or emotional pain on a child as retribution, or to force a certain reaction from them, and may include various forms of abuse [1,2,3]. Therefore, we distinguish discipline from abuse or coercion by describing it as non-violent discipline, rather than punishment.

*Discipline tools or options* are defined here as discrete, non-violent interventions that can be used to address a child or adolescent’s resistance, lack of co-operation, problem behavior or dysregulation, or to teach and support appropriate behavior [16]. *Tools* or *interventions* in this context describe individual practices such as distraction, prompting or time-out, rather than programs. The term *caregiver* is used to refer to the parent or any others who may be responsible for childcare and, therefore, discipline, for at least part of the day. This paper refers throughout to children and adolescents, but in most cases, the word *child* is used as shorthand to refer to both.

Training caregivers in non-violent discipline has been identified as an evidence-based approach to achieving the Sustainable Development Goals Target 16.2 of ending violence against children [24]. In addition, effective individual discipline tools could become “behavioral vaccines”, defined by Embry [25] as simple interventions which, if used widely, would substantially improve outcomes for a broad population at low cost and with minimal adverse effects. Embry [25] gives the example of the Good Behavior Game (GBG) [26], a classroom behavior management tool easily implemented by individual teachers. The GBG has been shown not only to be highly effective at reducing aggressive, oppositional and disruptive classroom behavior [27,28,29], but to have long term effects significantly reducing substance abuse, antisocial behavior and mental health problems by adolescence and young adulthood [30,31,32,33].

A wide range of non-violent discipline tools have been found and described [16,34,35]. Examples are improving communication, emotion coaching, adding structure, giving choices, increasing student opportunities to respond, modelling, social stories, goal setting, monitoring, daily behavior report card systems, prompting, praise, reward, group contingencies, time-out, problem-solving together, teaching replacement behaviors, and restorative interventions, which guide perpetrators to apologize and make amends [16,34,35]. Descriptions of a number of these tools can be found in the tables below. While there is a growing wealth of evidence of positive outcomes associated with their use, research on individual discipline options also shows that, like any intervention, they are not equally effective in all situations [16,34]. It is important to understand factors that influence the effectiveness of these tools, especially since caregivers and teachers who try non-violent discipline tools but find them ineffective, may abandon them and return to harsh methods such as corporal punishment. In this paper, we seek to deepen understanding of non-violent discipline by exploring attunement as a mechanism in the effectiveness of non-violent discipline tools.

### 1.2. Attunement

Attachment theory proposes that secure attachment with a primary caregiver is fundamental to human wellbeing [36,37,38,39]. Insecure or disorganized attachment to parents has been found to be significantly linked to unempathetic, aggressive, violent, and delinquent behavior [40,41,42,43]. It would thus be logical to target attachment in the prevention of problem behavior. It would also be important that discipline does not in any way damage the attachment bond [17], as could happen if harsh methods undermined the child’s sense of safety with their parents.

The term *attunement* was coined by Mary Ainsworth [36] to describe the parenting behavior she identified as central to the development of secure attachment: sensitive responsiveness [44]. Sensitivity described whether the mother noticed the baby’s signals, including subtle ones. Responsiveness described how her responses fit or matched appropriately with those signals. Attuned responses are “in tune with” [44] (p. 30) the child’s needs and inner state and can take many forms, such as emotional accessibility, physical comfort especially when the child is distressed, appropriate responses to states such as hunger, cold or overstimulation, acceptance and co-operation with the child’s desires and rhythms, and support for exploration. Misattunement could involve the mother being unresponsive, intrusive, or responding in some other way that does not fit with the child’s needs and signals [36,44]. 

Ainsworth observed that attunement built attachment security [36,37]. Subsequent research has confirmed that well-synchronized mother–infant interactions, in which maternal responses fit with the child’s signals, predict secure attachment [45,46], while regular misattunement can disrupt attachment, leading to various forms of insecure attachment [44,47]. Further, interventions effective in enhancing parental sensitivity have been shown to enhance attachment security [48]. 

Child self-regulation, which includes self-control and behavioral and emotion regulation, is an important goal and outcome of effective discipline [17]. Development of self-regulation has been found to be critical to success and well-being later in life, while low self-regulation has been significantly linked to violent and delinquent behavior, substance abuse and other negative outcomes such as poor financial management [49,50,51]. Aside from secure attachment, attunement has also been found to be significantly associated with child-self-regulation [52]. 

Considering the vital role of attunement in the development of self-regulation and secure attachment, and the vital role of each of these in the prevention of violent and delinquent behavior [42,50,51] and in the wellbeing of any person [39,50,51], it makes sense to investigate the idea that, for the best results in discipline (which is also aimed at preventing problem behavior and promoting self-regulation), adults should respond with attunement to the signals sent by the child through their behavior. 

*Attunement* in the context of discipline is defined, in this paper, as the matching or fit of disciplinary responses to the child’s needs and behavioral signals. An attuned response is thus one that fits the child’s signals in a way that best addresses the child’s needs and the function of their behavior. Disciplinary responses that match the child’s signals but do not match the child’s needs would not qualify as attuned; for example, if an adult accurately read behavior as signaling child fear, an attuned response would be reassurance or protection, not to use that fear to control the child. The former is in tune with the child’s needs, while the latter exploits the child’s emotion to serve an adult agenda. Since attunement has two parts: (1) understanding or accurate reading of needs and signals; (2) matching those needs and signals with an appropriate response, as a shorthand, *understanding* will be used for the first part and *matching* for the second part. 

Not all people responsible for discipline, for example, teachers, are attachment figures for the child, and they do not need to be. Beyond the parent–child relationship, the concept of attunement is also applied in counselling, with good counselor–client attunement predicting better therapeutic outcomes [53,54]. Attunement is considered to be particularly important in interventions for trauma, because focused attunement with another person has been found to shift people out of disorganized and fearful states [55]. Simply put, attunement seems to help people feel safer. This could be important in relation to the question of appropriate discipline for children with trauma histories and symptoms [17].

In classroom management research, a concept arguably synonymous with attunement emerged. Kounin [56] found that a characteristic he termed “withitness” was a better predictor of teacher efficacy than use of any particular discipline or classroom management skill. “Withitness” described the teacher’s constant awareness of what is happening in the classroom, and their ability to read and respond appropriately and promptly to the needs of the students. This definition of “withitness” could just as well be a definition of attunement, describing sensitive responsiveness on the part of the teacher, understanding the needs and signals of the students, and matching these with appropriate responses. Children taught by teachers showing this characteristic perform significantly better academically and behaviorally [56]. The importance of “withitness”, in relation to challenging classroom behavior, adds weight to the idea that attunement would be a logical focus in research on discipline, at school as well as in the home.

To the best of our knowledge, this has not been looked at before in a review. This makes sense when we consider that, although there are some exceptions such as collaborative problem-solving [57,58] and restorative justice interventions (e.g., [59,60]), most discipline tools were developed in the context of behaviorism [23] or applied behavior analysis (ABA) [16,61]. Research in the tradition of behaviorism or ABA only addresses observable behavior [62], not cognition, emotion, relational disconnection and repair, attachment bonds or internalizing behavior. While this exclusive focus on behavior has certainly resulted in more objectivity and scientific validity, it is also limiting, leaving a large gap in the research when it comes to the emotional and relational aspects of discipline. 

Since the majority of research available on behavior management tools does not address relationships or attachment, it would be unrealistic to expect to find evidence in reviews of such studies on the effects of attuned discipline on attachment security. It is possible, however, to use a review method to look at whether there is evidence suggesting that attunement, though of course not described using that term, could have an impact on the effectiveness of discipline tools. For example, in ABA, function-based interventions are interventions based on functional behavior assessment (FBA) or functional analysis. These are processes aimed at identifying the function of a problem behavior, i.e., the purpose it serves for the child [63]. Functions include reward (the problem behavior gets the child something they want), escape (the problem behavior gets them out of doing something they do not want to do) or automatic reinforcement (the problem behavior is enjoyable in itself) [63,64]. Once the function of the behavior is determined, an intervention is designed which addresses that function. This process of understanding what is underlying the behavior and then matching the intervention to the underlying need could be seen as a kind of attunement. A relatively consistent finding in reviews and meta-analyses of behavioral interventions is that interventions based on functional assessments are more effective in addressing challenging behavior than those that are not [65,66,67].

## 2. Method

A systematic overview of evidence on non-violent discipline options was conducted, the results of which have been published in a separate paper [16]. The protocol for this overview was based on the approach taken by the Cochrane Handbook for Systematic Reviews of Interventions [68] and other relevant sources [69,70,71,72], and approved by a review committee in the Department of Psychology at the University of Cape Town [16]. A summary of the method and search process used for the overview can be found in the Appendix A and a PRISMA flow diagram in Appendix A. Methodological quality was assessed using the AMSTAR checklist. A table showing AMSTAR scores of all reviews relevant to this paper can be found in Appendix A. The overview identified and described available evidence on a large number of non-violent discipline tools, drawing on data from 223 included systematic reviews. 

A further goal of the overview was to examine the evidence found in these reviews through the lens of attunement, to see if understanding the child and matching interventions to their needs and signals appeared to play any role in the use and effectiveness of the discipline tools. This is the focus of the current paper. Since attunement is primarily an attachment term, not a term used in behavioral or classroom management literature, it proved unproductive as a search term in relation to discipline tools. Instead, once individual non-violent discipline interventions had been identified from the included systematic reviews, these reviews were also examined for information relating to differential effectiveness of interventions or to the kind of understanding and matching that could be classed as attunement.

Data falling into any of the following 3 categories was considered relevant to this study: outcomes showing differential effectiveness of interventions across different subgroups of children or according to how the tool was used; outcomes of interventions which inherently involve the kind of understanding and matching that could be classed as attunement; and outcomes of tailored interventions, in other words, interventions custom-made to match the needs and signals of specific children. Further detail on these categories is provided in Section 3. Any reviews providing information in these categories were included. Data extracted from them are narratively summarized in the tables below.

## 3. Results

As expected, none of the reviews examined attunement directly. Out of the 223 systematic reviews screened, 114 were excluded, not offering any data relating to attunement [27,29,30,31,32,33,73,74,75,76,77,78,79,80,81,82,83,84,85,86,87,88,89,90,91,92,93,94,95,96,97,98,99,100,101,102,103,104,105,106,107,108,109,110,111,112,113,114,115,116,117,118,119,120,121,122,123,124,125,126,127,128,129,130,131,132,133,134,135,136,137,138,139,140,141,142,143,144,145,146,147,148,149,150,151,152,153,154,155,156,157,158,159,160,161,162,163,164,165,166,167,168,169,170,171,172,173,174,175,176,177,178,179,180], while 109 were included [28,34,57,59,60,64,181,182,183,184,185,186,187,188,189,190,191,192,193,194,195,196,197,198,199,200,201,202,203,204,205,206,207,208,209,210,211,212,213,214,215,216,217,218,219,220,221,222,223,224,225,226,227,228,229,230,231,232,233,234,235,236,237,238,239,240,241,242,243,244,245,246,247,248,249,250,251,252,253,254,255,256,257,258,259,260,261,262,263,264,265,266,267,268,269,270,271,272,273,274,275,276,277,278,279,280,281,282,283]. Data relating to attunement in any of the 3 categories described above were extracted from these included reviews. Findings are presented here. Reviews varied widely in quality, as can be seen by their AMSTAR scores in the Appendix A.

### 3.1. Aspects of Attunement Highlighted by Differential Effectiveness of Interventions 

Rather than thinking of a discipline tool as effective or ineffective in itself, or of one tool as generally better than another, differential effectiveness of an intervention across subgroups of children raises questions about whether that tool, or whether the way it was used, matched with the needs of the children it was used for [284,285]. Therefore, evidence that interventions work better for some than for others, or work better used in a certain way, can indicate areas in which attunement is needed. A number of reviews showed such evidence, highlighting several important aspects of attunement, such as appropriate match of intervention with child needs and abilities, developmental level or function of behavior. Table 1 summarizes this evidence. These examples illustrate that behavioral interventions can be very helpful for some children, but not very effective for others, depending on the child’s particular needs and abilities. This suggests that these tools should not be considered effective or good in themselves, but rather according to their fit with the needs of the child. 

### 3.2. Outcomes of Interventions Which Inherently Involve the Kind of Understanding and Matching That Could Be Classed as Attunement 

In contrast to the misattunement of harsh, coercive and punitive approaches to discipline, a number of non-violent discipline tools inherently involve some level of attunement. In order to use these tools, understanding of the child is necessary, as they need to be matched to the child’s area of difficulty, level of ability and their usual responses. For example, using behavioral momentum requires knowledge (understanding) of which tasks are easier for, or preferred by, the child, so that requests can be sequenced accordingly (matching). Any individualized intervention, for example, a social narrative, also requires a process of understanding and matching, because the intervention is tailored to match the needs and behavior of a specific child. The same can be said for any function-based interventions such as functional communication training, as these are designed to match the function challenging behavior usually serves for a specific child (e.g., attention or escape). Thus, a number of evidence-supported interventions inherently involve the kind of understanding and matching that could be classed as attunement. Table 2 describes these interventions, shows how they involve understanding and matching, and summarizes their outcomes. Since understanding and matching are integrally involved in the use of all of these interventions, it could be argued that their use demonstrates caregiver attunement. The positive outcomes found for them would, therefore, also suggest positive outcomes for attunement in discipline.

### 3.3. Outcomes of Tailored Interventions

Rather than being used in much the same way with any child, tailored interventions are interventions which are custom-made to match the needs and signals of specific children. While it is possible that a general intervention could match a particular child’s needs, a tailored or individualized intervention is specifically designed to do so. Function-based interventions are one example of this. As already mentioned, previous research has shown that function-based interventions are more likely to be effective in addressing challenging behavior than interventions that are not function-based [65,66,67]. This was confirmed in the current research, with findings from a number of included reviews providing evidence that function-based interventions and other forms of tailoring are associated with greater intervention effectiveness and other positive results. Table 3 summarizes this evidence. The superior outcomes of tailored interventions suggest the importance of attunement in the effectiveness of behavioral interventions.

### 3.4. Interventions Not Showing Evidence Related to Attunement 

Despite the fact that none of the included reviews specifically addressed attunement, some evidence was found relating to attunement in reviews on most of the discipline tools, as seen in the tables above. There were a few tools, however, for which reviewed evidence did not show evidence relating to attunement. These included certain aspects of structure, such as rules and policies, modelling, preparation, reprimands, and goal setting. One review on goal setting [189] mentioned that around half of the studies used individualized goals but none compared the effects of individualized goals versus generalized goals. This example illustrates the possibility that these few exceptions exist only because the reviews were not focused on attunement and, therefore, did not seek or present evidence that could have related to attunement. It is very possible that evidence would be found for attunement in the use of each of these interventions if it were sought. 

### 3.5. Interventions Likely to Improve Attunement 

A number of tools show potential to improve attunement. Anything that involves observation, monitoring, listening to a child, or anything that involves effort to increase knowledge and understanding of the child, could improve attunement if the adult is able to use their increased understanding to respond appropriately to the child’s needs. For example, attunement to the child could improve with good parent–child communication. The same could be said for interventions such as parental monitoring, playground supervision, use of a daily report card, or increased teacher-directed opportunities to respond (OTR), problem-solving together or restorative justice interventions. Each of these interventions could increase adult awareness and understanding of the child. There are also interventions which would improve adult understanding of the child by improving the child’s ability to send appropriate and readable signals, such as augmentative and alternative communication (AAC), functional communication training (FCT) and picture exchange communication system (PECS). Adults who understand children better should be better able to match their responses to child needs. 

## 4. Discussion

The above results highlight important aspects of attunement which could influence the effectiveness of disciplinary interventions. They show positive outcomes of a number of tools which inherently involve attunement, and superior outcomes associated with understanding and matching child needs through tailored as opposed to general interventions. Taken together, these results support the idea that attunement in discipline should be further investigated. 

It is possible that a toolkit of discipline options would be useful in scaffolding child development to the extent that caregivers are able to use those tools with attunement. From this perspective, individual tools such as reward, time-out or active listening would not be considered to be good skills in themselves but would only be good if they matched the needs and signals of the child at the time they were used. For instance, time-out could be a good skill to address aggression or non-compliance on the part of the child [197,219,230,241,251], but a bad skill to use if the child is experiencing fear or panic and actually needs reassurance or comfort [17,44]. Active listening may be a good skill to use if a child seems upset [288] but would not be the first choice if the child is unsafe, for example, if they are upset because the parent wants them to climb down from the 7th floor balcony railing, or stop running towards the middle of a busy road. Manual restraint in the form of taking the child off the railing or holding their hand at the roadside could be appropriate in the latter two cases, while there are plenty of other situations in which restraint is not called for and would be intrusive [289]. 

Beyond which skill to use, the above evidence (especially that presented in Table 1) suggests other important areas of attunement, such as whether intervention is necessary and constructive at all [202], whether it is appropriate [222,275], whether it is having the desired effect [251], and how much of an intervention is constructive. Regarding the latter point: at what point is a larger or longer dose of a particular skill unnecessary [197,225], at what point does a bigger dose undermine effects [252] or have negative effects [282]? 

For readers interested in more detail on what attuned discipline would look like in practice, further explanation and application through real examples can be found in the Appendix A. Three important areas are highlighted: understanding when and where a child needs behavioral support; choosing tools that would best fit the situation and needs of the child; and understanding when to stop intervening or to fade support.

An obvious limitation of this research is that attunement was not directly examined in any of the included reviews; however, it is hoped that highlighting this important area will attract research to it in future. To conduct research directly examining attunement in discipline, assessment tools will need to be developed and validated. The Dyadic Attunement Observation Schedule (DAOS) [290] is an observational measure currently used to score parent–child dyads videotaped during play interactions. A similar tool could be developed to generate attunement scores for adult–child dyads videotaped during disciplinary interactions. The Patient’s Experience of Attunement and Responsiveness (PEAR) Scale [54] is a self-report measure used immediately after a therapy session to assess the client’s experience of therapist attunement in that session. A similar tool could be developed for use after behavioral interventions to assess a child’s experience of adult attunement during the intervention. Scores could then be correlated with social and behavioral outcomes to show any moderating effects of attunement.

Observation suggests that securely attached children can become less secure and insecurely attached children more secure over time [37,38]. Future studies should examine the role of discipline, and especially attuned discipline, in these changes. The importance of discipline to attachment security is suggested by the fact that interventions such as Parent Child Interaction Therapy, which aim, and have been shown, to improve parent–child attunement and attachment, also rely on behavioral skills, coaching parents in the use of skills such as praise and time-out [44,291]. Also, parents who can rely on effective non-violent discipline tools may be less likely to become angry and frustrated to the point of using coercive and aggressive strategies which could undermine the child’s sense of safety with them [17]. 

Since attunement builds attachment security, it would be logical to predict that attuned or misattuned discipline would have implications for attachment security. One reason more research is needed on this, is that caregivers aware of the importance of attachment may unnecessarily avoid using certain non-violent discipline tools, as has been the case with time-out [17,197,292]. A reasonable hypothesis would be that attunement or misattunement in discipline would be more likely to impact attachment security than the use of any particular discipline tool. Research with a specific focus on this would not only deepen knowledge of attachment development processes, but also address concerns that may otherwise rob the toolkit of important and effective discipline options. See Appendix A for a more detailed discussion of misinformation and controversies surrounding certain tools [17,192,197,202,292,293,294,295,296,297,298].

Lastly, non-violent discipline tools have been shown to have important benefits beyond prevention of violence against children [16]. The fact that many of them inherently involve or have the potential to improve attunement adds to these benefits. 

## 5. Conclusions

A large range of evidence-supported non-violent discipline options for caregivers and teachers have been found and described, with outcomes not only showing effectiveness for challenging behavior, but many other benefits for child development [16]. This paper explored the potential role of attunement in the use of these discipline tools. Although none of the included reviews explicitly addressed attunement, evidence was found suggesting its importance in both the use and effectiveness of the reviewed interventions, highlighting the need for research directly investigating this. 

Attunement may be a better predictor of efficacy and social validity in discipline than use of any particular discipline tool, and measures need to be developed and validated to explore it further. This is a new frontier in behavioral research that could change the way discipline is approached. Rather than a “one tool fits all” approach, information made available on discipline tools could be accompanied by information explaining the need for attunement in their use. Caregivers and teachers could employ a model of “attunement plus options”, in which interventions are not considered good or bad in themselves, but rather evaluated according to their fit with the situation at hand, and the needs and signals of the child.

## Figures and Tables

**Table 1 ijerph-20-07187-t001:** Aspects of attunement highlighted by differential effectiveness of interventions.

Aspect of Attunement Highlighted	Examples
Does the intervention fit the child’s unique needs, sensitivities and preferences?	Children with Autism Spectrum Disorders (ASD) respond well to Visual Activity Schedules [221,223,231,268]. This intervention fits well with their preference for visual learning and strong need for predictability [221,231].
	Children with ASD respond well to video modelling [34,186,187,204,214,215,238,239,240]. This intervention fits well with their tendency to prefer visual learning [187]. One review noted that video self-modelling (VSM) alone yielded larger effects than VSM with reinforcement or as a component of a packaged intervention. Authors thought it likely that the other components increased social interaction with the interventionist, which would be more demanding for participants with ASD [238].
	Rewards have been found to be an important and effective intervention for children with ADHD. This intervention fits well with their heightened sensitivity to rewards compared to typical controls [233,236].
	Daily report cards and self-regulation interventions show large positive effects for children with ADHD [190,205,254,256,259]. These interventions fit well because frequent feedback about their behavior, a characteristic of both interventions, has been found to be a critical factor in their self-regulation [256]. It has been noted [256] that, since self-regulation is a deficit for children with ADHD, self-regulation interventions may be particularly important, although this deficit may also mean that some children are not yet capable of enough self-regulation to participate effectively in them [188]. These children may respond better at first to a daily report card, where feedback is given by an adult. Attunement would be needed to match the intervention to the child’s level of self-regulation.
Does the intervention fit with the child’s abilities?	When rewards were used to motivate children and adolescents with moderate to severe acquired brain injury (ABI) in rehabilitation settings, effects varied based on the severity of brain injury, with more severely injured participants showing less improvement with reward. This is in keeping with the finding that more severe injuries are likely to affect white matter structures known to be important in reward processing [271].
	A review on nocturnal enuresis [191] found that medication or enuresis alarms were often more effective than reward. It is possible that the children did not respond well to reward because they were not physically able to achieve the target behavior without the support of medication or an alarm.
Does the intervention fit with the child’s developmental level?	One review found that self-monitoring for children with ASD was more effective for older students. Authors suggest this could be attributed to more mature executive functions and thus greater developmental readiness for the intervention [200].
	In one study reviewed regarding toilet training for typically developing toddlers, 9 of 10 children who did not complete training were under 25 months. Review authors concluded that they may not have been old enough for toilet training. Thus, interventions such as prompting and reward, which are usually effective for toilet training, were ineffective because the children were not developmentally ready [278].
	Time-out shows the largest effects for boys under age 7 [273], suggesting greater need for this intervention for boys at this developmental stage.
Does the intervention fit with the function of the target behavior?	Functional communication training (FCT) has a strong evidence base [64,206,207,208,249]; however, if the function of a problem behavior is automatic reinforcement (i.e., it is rewarding in itself), it is less likely to respond to FCT [206], and extinction may not be possible. An antecedent intervention such as matched stimulation [209] or inhibitory stimulus control procedures [235] may be a better choice.
	The function of problem behavior has been found to play a key role in the effectiveness of check in check out (CICO), an intervention involving a daily report card. Strong effects were demonstrated for attention-maintained problem behavior, while, unless modified, it was ineffective for escape-maintained problem behavior [211,279].
	Time-out is usually effective for aggression [197,219,241,273]; however, interventions need to be matched to the function of the behavior [241], as, in one study, time-out reinforced aggression.
	Praise is such an important and effective intervention that sometimes students are taught to recruit it; however, for this to be effective, teacher attention needs to function as a reinforcer for the students [182].
Is the intervention necessary?	Rewards have been shown to undermine intrinsic motivation [202] but are usually effective where there is a lack of intrinsic motivation [192,202]. This suggests that if a child is not motivated to do something, reward would be useful, but if they are already motivated to do something, a reward would be unnecessary and may have undesirable effects.
How much is necessary/constructive?	Higher parental monitoring, including supervision and talking to parents of adolescents’ friends, was significantly associated with delay in age of first intercourse [282]; however, two studies showed that overcontrol was associated with earlier intercourse. This outcome suggests that parents need to attune to the amount of monitoring appropriate for their child, as too much or too little could have negative outcomes.
	Positive effects of choice diminish if too many choices (five or more) are given [252].
	Longer time-outs have not been found to add any benefits. Short time-outs (5 min or less) are usually enough [197].
	Restraint in the form of protective clothing or equipment has been used to reduce or prevent skin picking and eye gouging among adolescents with developmental disabilities. Continuous use of this equipment is not always necessary, however, as studies have shown that contingent use (e.g., gloves worn for a few minutes contingent on skin picking) can be more effective and easier to fade [225].
Is the intervention appropriate?	The use of extinction (planned ignoring) could be problematic for children with self-injurious behavior [275].
	Non-exclusionary time-outs have been shown to be effective for low-intensity or high-frequency inappropriate behaviors, but are not as appropriate for behaviors such as physical damage to self, property, or others [222]. For aggression, exclusionary time-outs have been shown to work well [273].
Is the intervention having the desired effect?	Praise is usually effective and beneficial [182,185,260,266,280]; however, two reviews showed that praise may not always be experienced as rewarding by children. Authors suggest that sometimes it may increase self-consciousness or be experienced as controlling [202,251].

**Table 2 ijerph-20-07187-t002:** Interventions which inherently involve the kind of understanding and matching that could be classed as attunement.

Intervention and Brief Description	How the Intervention Involves Attunement	Outcomes
*Antecedent interventions general:* Environmental modifications in which the events or circumstances precipitating the target behavior are altered. There are many different types.	Antecedent interventions require understanding of how the child is responding to their environment, in order to make appropriate adjustments to aspects of the environment that might impact on the child’s behavior (matching).	Reduction in problem behavior; increase in appropriate behavior. [34,205,208,226,243,247,259,264,267]
*Antecedent intervention: Behavioral momentum:* also referred to as high probability instruction/command/request sequence. Child is asked to complete series of 3 to 4 brief requests with high probability of compliance, just before a request with low probability of compliance. Thought to build momentum, increasing likelihood of compliance with low probability/preference requests.	Using behavioral momentum requires knowledge (understanding) of which tasks are easier for, or preferred by, each child (high probability of compliance), and which are not preferred, so that requests can be sequenced accordingly (matching).	Increased compliance [34,198,229,255]
*Antecedent intervention: Errorless compliance training:* Allowing child to demonstrate compliance at higher-probability requests, before systematically introducing lower and lower-probability requests.	Using errorless compliance training requires knowledge (understanding) of which tasks are easier for, or preferred by, each child, so that requests can be sequenced accordingly (matching).	Increased compliance (initiation and completion). [255]
*Antecedent intervention: Modifying task difficulty:* difficulty of a task is modified to lower the chance of escape or avoidance-motivated behavior.	Modifying task difficulty requires understanding of what the child finds difficult, in order to match task difficulty to child skill level.	Reduction in escape-maintained problem behaviors (e.g., challenging, destructive, aggressive, disruptive, noncompliant or off-task). [226,277]
*Antecedent intervention: Non-contingent reinforcement (NCR):* Reinforcement is added to the environment without the participant needing to earn it. Sometimes referred to as environmental enrichment, matched stimulation or time in.	NCR involves identifying the function of a problem behavior (understanding), such as automatic reinforcement, so that similar stimulation can be freely offered in the environment (matching).	Decrease in behavior maintained by automatic reinforcement such as self-injury, verbal or motor stereotypy and pica. Time-in associated with increased compliance. [209,241,255]
*Antecedent intervention: Preference/interest:* Interests or preferences of students are incorporated into required academic tasks.	Using preference involves ascertaining what the child prefers or finds interesting (understanding), in order to improve the match between task and interest.	Improvement in student behavior and academic performance. [246]
*Antecedent intervention: Social narratives:* Short, simple, individualized stories, usually with text and pictures, composed to help a child learn appropriate behavior in a specific social situation. Often used for children with ASD, has also been used for others with and without disabilities.	Social narratives are usually individualized. Knowledge is required (understanding) of what a child struggles with, and what appropriate behaviors they need to learn, in order to custom-design (match) a story for them, targeting specific behaviors in a specific situation.	Increase in appropriate behavior (e.g., social skills, communication; academic skills; adaptive skills), decrease in challenging behavior (e.g., aggression, disruptive behaviors). [34,220,227,244,258,269,283]
*Communication: caregiver-child:* Characteristics of good parent—child communication includes warmth, openness, respect, child disclosure, and talking about emotions.	Appropriate matching of adult responses to child signals is needed in communication, to reassure the child that they are understood and safe to disclose further.	Delayed sexual initiation and increased responsible sexual behavior; preventing or reducing adolescent substance use. Less delinquency: weak association for good communication, strong association for child disclosure. [195,196,213,224,248,261,270,282]
*Communication: emotional communication skills:* Caregivers’ *emotion socialization behaviors (ESBs)*: reactions to emotions, discussion of emotions, emotion coaching. Positive ESBs include being aware of low intensity emotion, supportive of emotional expression, and using emotions as opportunities for intimacy and teaching. May also include elaborative reminiscing, in which caregivers discuss past events with their child, acknowledging and validating the emotions experienced. Questions are asked about, or references made to emotions, and emotions are labelled, discussed and validated.	Emotional communication requires that the adult must read or understand child emotions and match them with accurate reflection, so that the child can learn about their emotions and develop language to express them.	Decreased likelihood of child conduct problems: (antisocial behavior; non-compliance, aggression, disruptive, defiant or oppositional behavior, or symptoms of DSM-IV/V disruptive behavior disorders); Improved parenting behaviors and skills. [217,219]
*Communication:* Dental or hospital staff *empathic communication*; listening; providing relevant information.	Any empathic communication requires understanding, so that responses and reflections will match what the child is communicating.	Reduced child fear-related behaviors; increased co-operation; improved child/adolescent hospital experience. [265,281]
*Communication: Functional communication Training (FCT):* Child is taught an appropriate communicative response to replace a problem behavior.	All FCT is function-based. Understanding (through functional analysis) is essential, as the communicative responses taught are individualized (matching), based on the function of the challenging behavior and the participant’s communication abilities. Without identifying the function, the challenging behavior serves for the child, an appropriate alternative response serving the same purpose cannot be devised and taught.	Decreased challenging behavior (e.g., aggressive, disruptive, destructive; self-injurious). [64,206,207,208,249]
*Extinction:* Once the function of a challenging behavior is identified, the reward is withdrawn, e.g., the reward of attention is withdrawn by ignoring the behavior (planned ignoring). Escape extinction involves not allowing the child to escape from the thing they don’t want to do, through tantrums or other challenging behavior.	Extinction relies on accurately identifying the function of the target behavior (understanding), so that the consequences reinforcing it can be withdrawn (matching).	Decrease in challenging behavior and increase in appropriate behavior in school and other contexts. Effective for bedtime problems and night waking. Escape extinction was effective for food selectivity and food refusal. Caution: An initial increase in the challenging behavior (an “extinction burst”) often occurs before the behavior is reduced. It is recommended that extinction should not be used in isolation, but with other interventions, such as teaching and reinforcing appropriate replacement behaviors. [34,208,245,264,266]
*Feedback on behavior: Daily report cards:* Reports on which students receive teacher feedback on target behaviors after every lesson. Usually used for students who frequently engage in off-task, disruptive or inappropriate behavior, and have not responded sufficiently to universal interventions which work for the rest of the class. There are also other forms of *performance feedback*, in which students are provided with data (e.g., charts, graphs, reports) systematically tracking their performance in target classroom behaviors/physical activity.	Target behaviors on Daily Report cards are usually individualized (matching), according to knowledge of the child’s usual behavior difficulties (understanding). Any intervention involving performance feedback potentially involves attunement because the child’s behavior must be closely observed, and the feedback given must accurately match child performance.	Decrease in challenging, disruptive and ADHD-type behavior; increase in appropriate behavior, academic achievement, school engagement and completion; improvements in social behavior. Other forms of performance feedback showed an increase in appropriate, prosocial and academic behaviors; decrease in inappropriate behavior; decrease in classroom transition times; short-term increases in physical activity. [199,205,211,216,254,259,263,266,272,279]
*Video modelling:* uses videos to provide a model of the target behavior/skill. There are different types. With video self-modelling, the child is recorded successfully performing the target behavior, with mistakes, problem behavior and adult prompts edited out.	Video self-modelling is an individualized intervention (matched). Videos are tailor-made to address each child’s specific target behaviors.	Effective for teaching appropriate behavior and skills. Reduction in challenging behavior. Particularly effective for children and adolescents with ASD. [34,183,186,187,194,212,215,238]
*Problem-solving together: Student participation* in decision making (e.g., re class rules or school problems): discussing, brainstorming, choosing and implementing solutions. *Collaborative problem-solving* approach: Adult attempts to solve a problem collaboratively with the child: adult explores child’s concerns about the problem; adult states their concern; adult and child brainstorm solutions that address both their concerns; child is given the first opportunity to generate a solution; no solutions are dismissed outright; adult helps child to think through whether each solution addresses both of their concerns and whether it is realistic and feasible; they agree on a solution, implement it and return to discuss whether it was successful. If not, they discuss further and try another solution until they find one that works.	Problem-solving with children involves listening to children’s views, concerns and suggestions (understanding), and finding solutions that fit those (matching). Collaborative problem-solving also involves attunement after deciding on a solution, as the adult must assess, together with the child, whether the solution worked (understanding) and, if not, keep trying solutions until an effective one is found (matching).	Student participation: qualitative results: increase in satisfaction, motivation, ownership, skills, competencies, knowledge, personal development, self-esteem, social status and democratic skills; improved student–adult relationships; improved school climate/culture; stronger sense of connection to school; higher perceptions of safety. A few studies reported negative effects: unmet expectations; negative feelings (e.g., not taken seriously; overwhelmed by responsibility). Collaborative problem-solving: Outpatient settings: improved parent–child relationships; reduction in oppositional behaviors, ADHD symptoms and parenting stress. Inpatient settings: dramatic reduction in use of restraint and locked-door seclusions; decrease in staff and patient injuries. School settings: reduction in disciplinary referrals and teacher stress. [57,210,242,274]
*Prompting:* Assisting or reminding a child to engage in a target behavior, usually as, or just before, they attempt the behavior. Prompts can be verbal, visual, gestural or physical and can be used systematically, in a hierarchy of least to most, or most to least intrusive prompts. Example of least to most prompting: proceeding, as needed, from visual to verbal to gestural to modelling to partial physical to full physical prompts.	Systematic prompting procedures use the least intrusive prompt necessary (e.g., a gesture rather than physical guidance) to prompt the desired behavior. The kind of prompt used is thus determined by the child’s behavior. When teaching a behavior, the interventionist works from most to least, and, when fading prompts, from least to most. The adult uses the child’s responses as a guide (understanding), only moving to the next level once the child manages (with most to least) or doesn’t manage (with least to most) the previous level (matching).	Increase in targeted behaviors such as toilet use, play skills, imitation skills, social skills, communication skills, academic skills; motor skills; vocational skills. Decrease in stereotypy for response redirection (a specialized form of prompting). [34,184,218,228,234,278]
*Differential reinforcement:* Desired behaviors are reinforced, while reinforcement for inappropriate behaviors is withheld or lessened. Several types, all involve making the problem behavior less reinforcing than the desired behavior.	Differential Reinforcement requires assessment of what variables are maintaining the problem behavior (understanding). Other components of intervention, such as extinction, teaching replacement behaviors or reinforcing other behaviors are based on this assessment (matching).	A well-researched skill, effective for wide range of target behaviors across different settings. Increase in appropriate behavior; decrease in inappropriate, disruptive, aggressive or self-injurious behavior. [34,205,225,237,241,253,257,266]
*Praise:* Adults express approval or admiration for appropriate behavior. With behavior specific praise, the adult gives verbal or written praise statements that explicitly describe the behavior being praised. The behavior would be something in the child’s control (e.g., effort) rather than out of their control (e.g., ability).	Behavior specific praise requires observation of the child’s behavior and matching positive feedback to that behavior.	Increased physical activity; healthier eating; appropriate classroom behavior (e.g., increases in on-task behavior, attention; correct responses; productivity; accuracy and academic performance). Decrease in inappropriate classroom behaviors. Students with and without disabilities taught to recruit praise received more praise, feedback and assistance and in turn showed increased task engagement, productivity and accuracy of work. Ineffective or showed mixed results for compliance. [182,185,230,251,260,266,280]
*Restorative Justice Interventions:* Restorative justice conferencing (RJC) includes victim-offender mediation (VOM) and the family group conference (FGC). VOM: mediator meets with victim and offender separately, to prepare them for a meeting with each other. Followed by a mediated session together, to speak about the crime and its effects, and decide together how best to repair the damage. FGCs: meeting between victim, offender, family members of both and a facilitator, to discuss the crime and its effects, and decide together on appropriate reparation.	Attunement to both perpetrator and victim is inherent in RJCs, both in the process of listening to all parties (understanding) and in matching of consequence to offense, since reparation fitting the specific crime and parties involved must be decided by all participants.	Mixed results regarding whether RJCs have effects on recidivism. One review found that behavioral program components such as behavioral modelling, behavior contracting, or parent training in behavioral skills (e.g., contingency management) had stronger prevention effects than restorative justice interventions. No suggestion that the restorative justice approach is less effective than traditional court processing. More sensitive measures than recidivism show greater victim satisfaction; slightly higher recognition of wrongdoing by offenders; less serious/harmful re-offenses. [59,60,201,232,250,262]
*Scripting and script fading:* A script is created for appropriate behavior in a specific situation, usually for participants with autism spectrum disorders (ASD). Scripts are practiced repeatedly, then used in real situations until successful, then systematically faded.	Scripting is individualized (matched), based on understanding of the child’s needs and difficulties with a specific situation. Fading progresses to each new step only when the child has mastered the current step (matching).	Increased social skills and communication; increased unscripted responses. [34,181]

**Table 3 ijerph-20-07187-t003:** Outcomes of tailored interventions.

Examples/Evidence	Aspect of Attunement Highlighted and Comments
Common components of effective interventions designed to improve parent–child communication about sex included developmental and/or cultural tailoring [270].	Awareness of (understanding) developmental level and cultural norms and matching the intervention to these.
Incorporating the child’s interests, motivation (through reinforcement), and targeting play skills that match the child’s developmental level were found to be common features of effective interventions [228] to teach play skills to children with ASD and PDD.	Understanding each child’s developmental level, interests and motivation and matching interventions to these.
In a review examining the good behavior game (GBG), authors highlighted the importance of making sure the rewards used are appealing to students. Of the four studies reporting use of preference assessments, three indicated large or moderate effects [29].	Understanding student preferences and matching rewards to these
Practitioners who employ preference assessments when using the antecedent intervention choice-making, are more likely to improve a student’s task engagement than those relying on choice-making alone [246].	Assessing (understanding) student preferences so that choices can be matched to these.
In medical settings, distraction interventions for pain, such as music or videos that were not tailored to the child’s preferences were more likely to produce higher effect sizes [193,203].	Understanding preference and matching distractions to these. Comment: At first glance, this may seem to be an exception to the rule, but attunement does not necessarily mean giving a child their preference. Better attunement in this case might be to identify what has the highest distraction value, which may mean something the child is less familiar with than their preferred music or video.
For children and adolescents with disabilities, augmentative and alternative communication (AAC) interventions based on functional behavior assessments (FBAs) had significantly larger effect sizes than those that did not use FBAs. Also, AAC with functional communication training (FCT) had significantly stronger effects in reducing challenging behavior. This may be attributed to the fact that FCT is function-based, utilizing FBA to inform intervention development [276].	Understanding the function of the target behavior for each child and matching the intervention to it.
For video modelling, custom-made videos were more effective than commercially available videos, and function-based videos more effective than non-function-based videos. One study compared reinforcement contingencies as part of the video intervention, and found that function-based consequences were more effective than non-function-based [187].	Understanding the function of the target behavior and custom making videos and rewards to match it.
Function-based antecedent interventions for stereotypy were slightly more effective than nonfunction-based interventions [247].	Understanding the function of the target behavior and matching antecedent interventions to it.
Goh and Bambara conducted a meta-analysis [208] of individualized Positive Behavior Support (PBS) interventions in school settings for children, with and without disabilities. Target behaviors were often severe, since in the PBS system, individualized interventions (Tier 3) are employed for children who have not responded to first (universal) or second tier (additional strategies, usually implemented in small groups) PBS interventions. All the interventions reviewed, (such as FCT, self-management, extinction, reinforcement, differential reinforcement or antecedent interventions) were effective. No significant differences in effect sizes were found between intervention types, intervention agents, settings, gender, grade or disability, although greater effect sizes were found where there had been team decision making. Authors attributed the lack of differences in results to the fact that all the interventions were highly individualized, based on FBA results, in other words they were carefully tailored for each participant. Authors concluded that functional assessment, rather than any specific skill, may be the “predominate influential variable governing intervention effects” [208] (p. 10).	Understanding the function of the target behavior and matching each intervention to this. Comment: Author’s conclusions echo Kounin’s finding, that “withitness” was a better predictor of teacher efficacy than the use of any particular discipline or classroom management skill [56]. The finding that team decision making improved results, may also relate to attunement, since input from more people working with each child should increase understanding of the child and their behavior, enabling better intervention choices.
Teachers and students rate function-based interventions highly [64].	Matching the function of the target behavior. Comment: This impact on social validity echoes the attachment research finding that when responses fit with the child’s signals, there is greater child satisfaction with the encounter [286,287] and increased security [36,45,46].

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
