# Peer review of "Using Non-Violent Discipline Tools: Evidence Suggesting the Importance of Attunement"

_ijerph, 2023, doi:10.3390/ijerph20247187_

Round 1

Reviewer 1 Report

Comments and Suggestions for Authors

I sincerely appreciate the authors' efforts to bring light to different types of non-violent discipline tools in the literature and to explore the role of attunement when evaluating the effectiveness of non-violent discipline tools. The authors conducted a systematic review and then provided a justification to study the concept of attunement when understanding the effects of parenting strategies. I believe that this manuscript can provide meaningful guidance for researchers, but there are several points of clarification that should be addressed to maximize the impact of this manuscript. Specifically, the main concerns lie in the author's conceptualization of categories of data related to attunement.  

Abstract:

1. I suggest keeping the abstract concise to the discussion of "attunement" and preventing adding related concepts (line 15) to keep the focus on the topic. 

Introduction:

1. I recommend that the authors reorganize their first and second paragraphs to improve the flow. As a reader, the first paragraph gave me the impression that this paper is going more in the direction of coercive and violent parenting. I suggest tying the importance of utilizing non-violent parenting at the end of the first paragraph to prompt the transition. The first sentence of the second paragraph (line 39 - 44) is very long and make it hard to read. I recommend breaking up long sentences into shorter ones throughout the manuscript to improve readability. 

2. I appreciate the in-depth explanation of the concept of "attunement." While concepts of "goodness of fit" and "withitness" are related to attunement, it would be helpful to bridge their relations more explicitly. Currently, it reads as if they are semi-related/standoff-ish concepts but tying all these concepts more closely to "attunement" would be helpful. If there is a struggle to explicitly link them together, I recommend removing these related concepts as there are enough justifications of why attunement is an important concept to focus on when understanding the effectiveness of non-violent discipline. 

3. I suggest authors reorganize and incorporate the definitions section throughout the discussion and shorten the discussion section. While having more information demonstrates some breadth of the topic, it is more helpful to keep it concise to the main topic. Currently, the introduction is very long and it can be shortened to the focus of the paper (e.g., non-violent discipline and attunement). It is more helpful to have the definitions earlier as it provided more context to the main variables and scope of the review. 

4. I recommend authors add a sentence somewhere referring to later tables that provide the definitions of the common types of non-violent. 

5. The use of "attunement as moderator" language seems inappropriate in this paper as there is no direct examination of moderating effects. "mechanism" or "influencing factor" may be more appropriate. Authors may include examining moderating effects as a future direction.

Methods

1. I suggest authors move the first paragraph from the results section (lines 258 - 266) into the methods section. The methods section right now does not provide details that pertain to this study, it still mostly reads as if I am reading Quil & Ward (2020). I recommend having a new PRISMA flow diagram that further outlines how 223 articles went to the remaining 122 articles.

2. The language and conceptualization for 4 categories of data related to attunement is confusing. It may be more helpful to keep it as close to the original definition that you provided - "Attunement in the context of discipline is defined as the matching or fit of disciplinary responses to the child’s needs and behavioral signals." For example, "outcomes of interventions which inherently involve the kind of understanding and matching that could be classed as attunement" can be summarized as "discipline strategies demonstrate caregiver's ability to understand and match with the child’s needs and behavioral signals". 

3. I really liked the heading "Aspect of Attunement Highlighted" as shown in Table 4. I think this should be the main driving point of the manuscript and the results section should be centered on the argument that all the discipline strategies involved some aspect of attunement. I recommend the authors reorganize the categories to focus on highlighting aspects of attunement (E.g., understanding, matching). 

3. I question the need to have a separate category for "effects of interventions likely to improve attunement". This category reflected some underlying assumptions that each of those studies specifically included  "matching" or "understanding" criteria as outcomes. Felt like this category us too speculative. For example, the authors noted that "PECS improves the child’s ability to send appropriate and readable signals. This would improve adult understanding of the child’s state, and in turn, increase the chance that adult responses would fit with child needs (matching)." Did these studies involving PECS explicitly show evidence of an increase in a child’s ability to send appropriate and readable signals? I argue that this category should be removed and combined with the first category since these sections demonstrate the authors' speculation of the mechanism of each non-violent discipline strategy -- you identified "matching" or "understanding components."

4. I find the category of "outcomes of tailored interventions" to be confusing and unclear. How is the concept of tailoring different from matching? For example, authors noted "tailor-making the intervention for a specific group", how is this different from matching interventions to the group's cultural and background norms? The concept of tailoring was not included in the initial definition of attunement, perhaps this needs to be added. Perhaps this issue can be resolved by having a more thoughtful category name, or by including a discussion on how tailoring and matching are similar vs different components of attunement. 

Discussion:

1. It is important to separate the content/concept vs examples to help readers better understand the idea presented by the authors. For example, in the sentence: "Active listening may be a good skill to use if a child seems upset [189] but should not be the first choice when a young child is upset because the parent wants them to climb down from the 7th floor balcony railing or stop running towards the middle of a busy road." the second part of the sentence does not reflect content/concept but an example. I think this sentence can be improved with "Active listening may be a good skill to use if a child seems upset in a safe environment [189] but should not be the first choice when a young child is upset in context of compromised safety (e.g., the parent wants them to climb down from the 7th-floor balcony railing or stop running towards the middle of a busy road)." Please check throughout the paper to make relevant edits to illustrate the concept vs example. 

2. While I agree with the authors that it is important to directly examine attunement when evaluating the effectiveness of non-violent strategies, discussions of how attunement should be measured in research settings are warranted. For example, behavioral observations/coding of how parents demonstrate "understanding" and "matching" etc. A validated tool is needed. 

3. The paragraph on why more research is needed on how discipline affects attachment security seems repetitive and redundant. This point has been emphasized throughout the paper. Perhaps summarize this and include it in the conclusion.

4. I really liked the materials provided in the supplementary material 5. I recommend authors shorten the current manuscript to make space for summary statements from the supplementary material 5. Adding this information to the discussion would improve the practicality of the paper.

Comments on the Quality of English Language

Good work so far! I recommend breaking long sentences into shorter sentences to improve readability. Be explicit when illustrating concepts vs examples to improve reader's comprehension of the subject.

Reviewer 2 Report

Comments and Suggestions for Authors

I congratulate the authors on the originality of their work. I would like to highlight the relevance of applying the concept of attunement in the context of behavioral hetero-regulation, as well as the innovative way in which this research was conceived and carried out.

The work is robust, and part of this robustness derives from the systematic review previously carried out on non-violent discipline options. The concept of attunement applied to the context of discipline has been clearly operationalized. The tables presented in the results section clarify the systematicity of the work carried out.

In the method section, the categories classifying the attunement-related data need to be better substantiated and justified. The reasons behind the choice of these categories are unclear. On the other hand, while the first category is operationalized considering the criteria that define the concept of attunement, the remaining categories are only tacitly operationalized. These 3 categories should be explicitly justified in the method section, considering the criteria underlying the concept of attunement.

Since “none of the included reviews specifically addressed attunement” as it is stated in section 4.5 of the manuscript, the first sentence of the discussion is written in a way that suggests that the results are more far-reaching than they actually are. Although I believe that this was not the authors' intention, the sentence should be rewritten in a more specific and parsimonious way.

Reviewer 3 Report

Comments and Suggestions for Authors

This is a very good manuscript for a systematic literature study, however, it can be optimized and improved in the following points:

1. In the introduction, the importance of conducting this study needs to be further strengthened and elaborated.

2. In the methodology section, in fact, nowadays systematic literature reviews and studies often use software such as Connected Papers for content relevance mining, however, this method was not used in this thesis, and the relevant reasons need to be explained.

3. In the results, there is too much content inside these tables, rather than being conducive to reading and understanding. It is recommended that the contents of all the tables be better categorized and streamlined so that they can be understood at a glance.

4. The discussion section should make the research shortcomings and future research directions clear.

5. The conclusion section is too bland and needs to be improved.

Round 2

Reviewer 1 Report

Comments and Suggestions for Authors

I reviewed the revised paper and the responses to each of my comments. I find that my concerns have been adequately addressed. However, I think that the authors should add a description of their results from the Methodological quality of reviews assessed using the AMSTAR checklist.

Reviewer 3 Report

Comments and Suggestions for Authors

This paper meets the criteria for publication.
